# Resporulation of Calcium Alginate Encapsulated *Metarhizium anisopliae* on Metham^®^-Fumigated Soil and Infectivity on Larvae of *Tenebrio molitor*

**DOI:** 10.3390/jof8101114

**Published:** 2022-10-21

**Authors:** Sudhan Shah, Gavin J. Ash, Bree A. L. Wilson

**Affiliations:** 1Centre for Crop Health, University of Southern Queensland, Toowoomba, QLD 4350, Australia; 2Department of Agriculture and Fisheries, Ecosciences Precinct, Dutton Park, QLD 4102, Australia

**Keywords:** *Metarhizium anisopliae*, entomopathogenic fungi, calcium alginate granules, biological insect control, soil fumigation, soil insects

## Abstract

*Metarhizium anisopliae* infects and kills a large range of insects and is a promising biocontrol agent to manage soil insects, such as wireworm in sweetpotato. The presence of other soil microbes, which exhibit competitive fungistasis, may inhibit the establishment of *M. anisopliae* in soil. Microbially depleted soil, for example, sterilized soil, has been shown to improve the resporulation of the fungus from nutrient-fortified *M. anisopliae.* Prior to planting, sweetpotato plant beds can be disinfected with fumigants, such as Metham^®^, to control soil-borne pests and weeds. Metham^®^ is a broad-spectrum soil microbial suppressant; however, its effect on *Metarhizium* spp. is unclear. In the research presented here, fungal resporulation was examined in Metham^®^-fumigated soil and the infectivity of the resulting granule sporulation was evaluated on mealworm, as a proxy for wireworm. The fungal granules grown on different soil treatments (fumigated, field and pasteurized soil) resporulated profusely (for example, 4.14 × 10^7^ (±2.17 × 10^6^) conidia per granule on fumigated soil), but the resporulation was not significantly different among the three soil treatments. However, the conidial germination of the resporulated granules on fumigated soil was >80%, which was significantly higher than those on pasteurized soil or field soil. The resporulated fungal granules were highly infective, causing 100% insect mortality 9 days after the inoculation, regardless of soil treatments. The results from this research show that the fungal granules applied to soils could be an infective inoculant in sweetpotato fields in conjunction with soil fumigation. Additional field studies are required to validate these results and to demonstrate integration with current farming practices.

## 1. Introduction

Root crops, such as sweetpotato, are vulnerable to various soil-borne pathogens and soil insect pests [1,2]. In Australia, root herbivores, particularly plant-parasitic nematodes (*Meloidogyne javanica* and *M. incognita*), wireworms (both *Elateridae* and *Tenebrionidae*) and sweetpotato weevils (*Cylas formicarius*) can cause substantial damage to sweetpotato roots [3,4,5]. This damage often leads to economic losses for growers as the produce is either rejected at the packing shed or is sold as a non-premium product. Therefore, chemical pesticides are often seen as being crucial for crop protection in Australia [6]; however, their use against root herbivores, such as wireworms in sweetpotato, does not always lead to reduced insect populations. Specifically, wireworms intensify their feeding damage up until the crop is harvested (preventing the use of chemicals with harmful residues), while the efficacy of pre-plant-applied soil insecticides is depleted over time. Therefore, the use of entomopathogenic fungi has been proposed as an additional tool to manage wireworms in an integrated pest management program for sweetpotato.

Entomopathogenic fungi, such as *Beauveria bassiana*, *Metarhizium anisopliae* and *Isaria fumosorosea,* have been frequently detected in soil from a diverse range of habitats and show great diversity due to their worldwide distribution [7]. Several studies have shown that *B. bassiana* is predominantly found in natural habitats, for example, forests, while the prevalence of *M. anisopliae* is concentrated in cultivated soils [7,8]. Considering the association of *M. anisopliae* in agriculture fields, irrespective of practices, such as tillage and farm inputs [9,10,11], many studies have evaluated the ability of *M. anisopliae* to infect soil insect pests [12,13,14]. Despite this, *M. anisopliae*-based management of insects is not reliable in field conditions [15,16], even though substantial insect mortality by *M. anisopliae* has been observed under laboratory conditions [17,18]. For this reason, fungal adaptation, especially in agroecosystems, has been viewed as a crucial attribute that needs to be evaluated for any entomopathogenic fungal (EPF) candidates before their field application. Some *Metarhizium* species, for example, *M. anisopliae*, *M. brunneum* and *M. robertsii,* demonstrate a strong association in the soil environment in terms of rhizosphere colonization [19,20] and endophytic establishment in plant roots, for example, in cassava [21] and in tomato [22]. Fungal–plant interactions, such as endophytism or rhizosphere competence, can confer crop protection against root herbivores [20] and even result in the suppression of root herbivores by altering the gene expression of the host plant [23]. Not surprisingly, the degree of crop protection conferred by EPF against soil insects relies on the density of fungal inoculum. For example, only the repetitive application of inoculum (*B. brongniartii* Sacc.) could achieve the required EPF density in the soil to cause mortality to the European cockchafer (*Melolontha* spp.) [24]. Entomopathogenic fungal inocula targeting the control of soil insects have been formulated as colonized grains, such as barley, millet, rice and maize [13,25,26]. These grains nutritionally support the growth of EPF propagules in soil, providing crop protection against soil insects [27,28]. The presence of insect and plant hosts in the soil can further improve the EPF growth and persistence in soil, but the saprophytic growth of EPF is important to maintain propagules at high enough levels to result in insect infection and rhizosphere colonization [29]. However, fungistasis initiated by indigenous soil microbes has often been implicated in hindering the growth of the target EPF in soil [30]. Fungistasis in the soil can be manipulated by reducing the microbes living in soil, for example, using soil sterilization, which can result in substantial growth from the applied EPF inoculum, leading to significantly greater insect mortality in comparison to untreated soil [31,32]. The practice of soil disinfection has been universally adopted to particularly control soil-borne pathogens in many crops. Methods for soil disinfestation can include both solarization and fumigation and these methods have been implicated in temporarily suppressing a broad range of soil microbes, especially spore-forming fungi [33,34].

The practice of soil fumigation with methyl bromide (CH_3_Br) was once prevalent globally as it was a highly effective nematicide, insecticide, fungicide and herbicide [35]. The global ban of methyl bromide stimulated the search for alternative fumigation options, including non-chemical methods, such as solarization, soil steaming, electromagnetic radiation and biofumigation [36,37]. The practice of biofumigation, applied alone or in combination with solarization, is broadly practiced in fields by incorporating the parts of glucosinate-containing plants, particularly brassica crops (for example, *Brassica napus*) into the soil [38,39]. Myrosinage, an endogenous enzyme found in brassica plants (e.g., *B. napus*), hydrolyses glucosinolates converting them to isothiocyanates (ITCs), which possess biocidal properties [40].

Metham sodium (MS) is a soil fumigant based on sodium N-methyldithiocarbamate, which converts to methylisothiocynate (MITC) upon contact with moist soil [41]. Methylisothiocynate is toxic to soil biota and has been widely used in agriculture as a suppressor of weed seeds (for example, *Portulaca oleracea* and *polygonum arsenastrum*) [42], plant-parasitic nematodes (for example, *Meloidogyne incognita* in cucumber) [43], plant pathogenic fungi (for example, *Verticillium dahlia*) [44] and soil insects (for example, soil-dwelling white-fringed beetle *Graphognathus leucoloma*) [45]. Mixed success of soil fumigation with metham sodium against phytopathogenic bacteria has been reported, for example, no efficacy against *Pseudomonas allicola* was observed in onion [46], whilst significant control of *Erwinia carotovora* was observed in summer squash in a field following metham fumigation [47]. Due to its control efficacy for a broad range of weeds, nematodes and fungal pathogens, metham sodium has been applied to manage the weeds and pests of multiple crops, including in the preparation of soil for sweetpotato nursery beds in Australia. The application of metham sodium to the soil also causes off-target effects to multiple soil microbes, including bacteria and fungi. The abundance and diversity of soil fungal and bacterial communities were significantly reduced in metham-sodium-fumigated soil, including the beneficial mycorrhizal fungi [48,49]. However, because methylisothiocynate rapidly degrades in the soil, some soil microbes, particularly actinobacteria and proteobacteria, can recolonize the soil quickly [50].

In our previous work, encapsulated *M. anisopliae* granules that were inoculated into heat-sterilized soil effectively resporulated, resulting in high mortality of model insects, larvae *T. molitor* under laboratory and glasshouse conditions [31]. To the best of our knowledge, there has not been a study that has examined *M. anisopliae* granules, particularly calcium-encapsulated granules and the resporulation and infectivity of these granules in metham-fumigated soil. Because metham fumigation has been used to suppress weeds, nematodes and fungal pathogens in the preparation of soil for sweetpotato nursery beds [3,6] and EPF have shown promise to be incorporated into integrated pest management programs to control insect pests [25,31], we sought to examine the effects of fumigation on the persistence, saprophytic competence and infectivity of *M. anisopliae* in soil. *Metarhizium anisopliae* was formulated with nutrient-additive fortification, referred to as fungal granules, while larval mealworms (*Tenebrio molitor*) were used as a model insect for soil insects, such as wireworm in sweetpotato.

## 2. Materials and Methods

### 2.1. Metarhizium anisopliae

*Metarhizium anisopliae* strain QS155 (accession number DAR 82480) was originally isolated from the soil at Mapuru, Northern Territory, and is maintained at the University of Southern Queensland and at the New South Wales Department of Primary Industries Herbarium [4]. Cultures of *M. anisopliae* were grown on Sabouraud Dextrose Agar amended with 1% Yeast extract (SDAY) (Merck KGaA, Darmstadt, Germany). To produce conidia, *M. anisopliae* isolate QS155 was grown on SDAY at 27 °C with a 12:12 h light and dark photoperiod for 21 days. The conidia were harvested using a sterile scalpel by gently scraping the colony and were dried in Petri dishes (Ø, 9 cm) in a biohazard cabinet (Esco class II BSC) for 2 h. The air-dried conidia were stored in sterile plastic 50 mL tubes and sealed with a lid at 5 °C for 7 days until the conidia were prepared for formulation.

Before all experiments, the conidia viability was assessed by inoculating 20 µL of a conidial suspension (10^6^ conidia per mL) over a thin layer of SDAY medium (1.5 × 1.5 × 0.5 cm) on a glass microscope slide, which was covered with a coverslip and placed inside a Petri dish (Ø, 9 cm) containing Whatman^®^ (Maidstone, United Kingdom) filter paper moistened with sterile-distilled water. The Petri dish was sealed with Parafilm^®^ (Neenah, WI, USA) and incubated at 27 °C with a 12:12 h dark and light photoperiod. Following 14 h of incubation, 200 conidia were assessed at ×400 using a compound microscope (Olympus, Model BX53, Melbourne, Australia). Only samples with >98% germination were used for further experimentation.

### 2.2. Preparation of M. anisopliae Granules

To prepare the calcium–alginate formulation, 2% (*w*/*v*) sodium alginate (Chem-Supply Pty Ltd., Gillman, SA, Australia) was dissolved in 0.05% sterile Tween^®^80 (VWR Chemicals, Radnor, PA, USA) in water; the resulting suspension was heated with continuous agitation for 30 min before being autoclaved at 121 °C for strictly 6 min, to prevent chemical denaturation of sodium alginate [51]. The conidia of *M. anisopliae* QS155 were mixed into the sodium alginate, in combination with nutritive additives: 20% *w*/*w* corn starch (Sigma-Aldrich, St. Louis, MI, USA) and 20% *w*/*w* compressed baker’s yeast (Lesaffre Australia Pacific Pty Ltd., Dandenong South, VIC, Australia) [31]. The above nutritive additives, which were autoclaved at 121 °C for 15 min, were suspended in the sterile sodium alginate solution and homogenized thoroughly using a stirrer for 10 min. Fresh conidia of *M. anisopliae* QS155 (1% *w*/*w*) were then added into the suspension and stirred using a stirring rod for 5 min. The homogenized suspension was immediately dripped into sterile 2% (*w*/*w*) calcium chloride solution (ICN Biomedicals Inc., Costa Mesa, CA, USA) using a syringe (Norm-Ject^®^, drain tube Ø = 4 mm, length = 10 mm). The droplets of the suspension remained immersed in the calcium chloride solution for 30 min with continuous agitation for complete gelatinization [50]. Granules were separated from the calcium chloride solution by collecting them on a sterile Buchner funnel. Granules were rinsed twice with sterile water before being dried for 14 h inside a laminar flow cabinet (Labec Laboratory Equipment, Marrickville, NSW, Australia) at room temperature (22–24 °C). The drying process removed 55% of moisture from the granules, which were then sealed in a 100 mL container and stored at 5 °C until the experiment commenced. These granules are referred to as CAG_Ma+Cs+By_ (Ø = 3.5 mm, weight 25 mg per granule, 9 × 10^6^ conidia per granule) henceforth where ‘CAG’ is defined as calcium alginate granule, ‘Ma’ is *M. anisopliae*, ‘Cs’ is corn starch and ‘By’ is baker’s yeast (Figure 1A). For the control treatment, calcium alginate granules only contained the nutritive additives (20% *w*/*w* corn starch and 20% *w*/*w* baker’s yeast) without *M. anisopliae* and are referred to as food granules (CAG_Cs+By_) (Figure 1B).

### 2.3. Collection of Soil for Experimentation

Soil (clay 60%; silt 20%; sand 20%; pH 6.6; EC 7 mS/m; Carbon 3.22%; Nitrogen 0.22%) was sourced from an agricultural field of the University of Southern Queensland, Toowoomba (GPS coordinate: 27°36′33′′ S, 151°55′55′′ E) in February 2021 (monthly mean minimum and maximum temperature 16.6 and 26.6 °C, respectively [52]. The field was planted with barley in winter but left fallow during summer. The soil was sampled across the field in a diagonal transect, where 6 samples were taken over 100 m. For each sample, the top 5 cm of the topsoil was cleared and soil was collected to a depth of 15 cm using a trowel. Soil samples were pooled (~6 kg weight) and immediately transported to the laboratory for further processing. The sample was homogenized, sieved (5 mm aperture) and air-dried for 12 h, before being transferred to a plastic bin (20 L) fastened with an air-tight lid and stored in a cool room (5 °C) until further use. This processed soil sample is referred to as ‘field soil’.

### 2.4. Soil Treatment: Fumigation, Pasteurization and Field Soil

In this experiment, the resporulation and infectivity potential of fungal granules was investigated on three soil treatments, fumigated, pasteurized and field soil. To fumigate the soil, transparent plastic containers (diameter 8.5 cm; height 9 cm) were filled with 150 g of field soil. All containers containing field soil were adjusted to 50% field capacity (16% initial moisture content) by applying 40 mL of ultra-pure Type 1 water (Milli-Q®, Merck, Bayswater, Australia) to each container 8 h before being treated. The containers were then treated with Metham soil fumigant (active ingredient 423 g/L metham as a sodium salt (source Nufarm Australia) at the rate of 600 mL/m^3^ soil using a pipette). Following the application of the Metham, an additional 5 mL of ultra-pure Type 1 water (Milli-Q) was added to each container using a syringe (volume 60 ^cc^/mL; Terumo^®^ Syringe, Macquarie Park, Australia) and the lids were sealed tightly to contain the volatile fumes. Seven days after the fumigant was applied, the lids were loosened by half a turn to allow for gas release. The process of soil fumigation was performed inside a fume hood (Lab systems, Product for Science and Life) at room temperature: 22 °C and 79% RH for 19 days.

For the soil pasteurization treatment, 2 kg of field soil (16% initial moisture content) was placed into a double-layered aluminum tray (20 cm × 15 cm × 5 cm), moistened with 400 mL of ultra-pure Type 1 water (Milli-Q) and wrapped with a double layer of aluminum foil to prevent heat loss and cross-contamination. The soil was heated to 80 °C for 40 min in an oven (Memmert GmbH + Co. KG, Australia) The pasteurized soil was dispensed into plastic containers (diameter 8.5 cm; height 9 cm) containing 150 g of pasteurized soil per container and adjusted to 50% field capacity at 8 h before being treated. Pasteurized soils were treated with 5 mL of ultra-pure Type 1 water (Milli-Q). All the procedures were carried out at the same time as the fumigated soil, but in a separate fume hood at 22 °C and 79% RH to avoid cross-contamination of any volatiles. The containers were treated as described for the fumigated soil.

For the field soil treatment, the same containers were filled with 150 g of non-sterile field soil per container (diameter 8.5 cm, height 9 cm) and the soil moisture was adjusted to 50% field capacity 8 h before the ‘treatment’ (placement in the fume hood). The soil was treated with 5 mL of ultra-pure Type 1 water (Milli-Q) only. All procedures were carried out at the same time with the fumigated and pasteurized soil. All containers containing pasteurized and field soils were placed separately from fumigated soil into the fume hood during the treatment period to prevent cross-contamination. The other procedures performed were the same as for the fumigated and pasteurized soils.

### 2.5. Cultivation of Soil Microbes

Prior to treating the prepared soils with *M. anisopliae* granules, the colony-forming units (CFUs) of background soil microbes were enumerated for each soil type to determine the efficacy of the soil treatments on the presence of soil microbes. For the fumigated soil, CFUs were calculated from surface soil where the fumigant was applied (Fumigant (1)) and from the homogenized fumigated soil after the fumigation was completed (Fumigant (2)). For each soil type (field soil, pasteurized soil or two fumigated soils), 1 g of soil was suspended in a 20 mL plastic tube containing 9 mL of sterile water and was agitated for 5 min using a vortex. A dilution series from 10^−2^ to 10^−8^ was made for each soil type. Serial dilutions of 10^−2^ to 10^−4^ were assigned to pasteurized soil and fumigated soil, whereas dilutions ranging from 10^−4^ to 10^−8^ were used for field soil samples. For each soil dilution, 100 µL of suspension was plated out onto triplicate Petri dishes (Ø, 90 mm) containing potato dextrose agar (Merck KGaA, Germany) or nutrient agar (Merck KGaA, Germany) using a pipette. Petri dishes were sealed with Parafilm^®^ and incubated in a controlled temperature room at 25 °C with a 12:12 day and light photoperiod for 7 d.

To determine the presence of indigenous EPF from field soil, a plastic container (4 cm diameter; 8 cm height) was filled with 50 g of moist soil sourced from either field soil or pasteurized soil or fumigated soil prior to the addition of *M. anisopliae* granules. To each container, ten larval mealworms were added and the containers were sealed with a perforated lid and incubated in a controlled temperature room at 25 °C and a 12:12 day and light photoperiod. If any dead mealworms were found, they were surfaced-sterilized with 100% ethanol, rinsed in distilled water and placed into a moist chamber (90 cm Petri dish lined with a moist sterile Whatman^®^ paper). The Petri dishes were incubated as above to stimulate EPF sporulation.

### 2.6. Fungal Inoculation and Experimental Design

After 19 days, individual containers containing fumigated, pasteurized soil or field soil received 5 mL of sterile water before being inoculated with *M. anisopliae* (CAG_Ma+Cs+By_) (Figure 1A) or food (CAG_Cs+By_) granules (Figure 1B). Prepared *M. anisopliae* granules were placed on the soil surface at a rate of 4 × 10^8^ conidia g^−1^ soil equivalent to 1 g (~60 pieces) of granules. Food granules without conidia were inoculated into the respective containers with soil as the control treatment. A Petri dish containing 3% water agar was inoculated with either ten fungal granules or food granules for quality control (to assess for resporulation and the presence of contaminants), sealed with Parafilm^®^ and incubated in a plant growth chamber (CONVIRON^®^ CMP6010, Melbourne, Australia) at 25 °C, 80% RH and a 12/12 h day and night photoperiod. After inoculation, the containers were loosely sealed with a lid and transferred to the same plant growth chamber and incubated at 25 °C, 80% relative humidity (RH), with a 12:12 h dark and light photoperiod. There were six replicates per treatment and containers were arranged in a randomized complete block design (RCBD).

### 2.7. Resporulation Assessment

To determine the amount of resporulation from the *M. anisopliae* granules at 10 days post-incubation, three *M. anisopliae* granules (CAG_Ma+Cs+By_) or food granules (CAG_Cs+By_) were randomly removed from each container using sterile tweezers (Figure 2A–C). The conidial resporulation from the granules was determined by inserting an individual *M. anisopliae* granule into a tube (2 mL) containing 1 mL of sterile 0.05% Tween^®^ 80 solution. The tubes were agitated using a vortex for 2 min to dislodge the conidia from the granule.

Conidia from the suspension were enumerated using a haemocytometer (Neubauer improved double net ruling, ProSciTech Pty Ltd., Kirwan, Australia) at ×400 magnification (Olympus, Model BX53, Melbourne, Australia). The conidial germination was assessed by inoculating 20 µL of each suspension onto a glass slide with a thin layer of SDAY agar (1.5 × 1.5 × 0.5 cm). A coverslip was placed on the agar and each glass slide was placed in a Petri dish (Ø, 9 cm) lined with a moist Whatman^®^ filter sealed with Parafilm^®^ and incubated at 27 °C and a 12:12 h dark and light photoperiod. For each slide, at 12 h, 14 h, 16 h and 24 h post-incubation, the germination of 200 conidia was assessed at ×400 magnification using a compound microscope (Olympus, Model BX53, Melbourne, Australia). Conidia with a germ tube twice as long as the width of spores were considered to be germinated [53]. During the germination test, the length of the germ tube from the germinated conidia (20 conidia per replicate) was also measured.

### 2.8. Infectivity Assessment

At 10 days post-incubation, the infectivity of the remaining fungal granules (calculated at ~1.43 × 10^7^ conidia g^−1^ soil based on the resporulation of fungal granules following the incubation) was evaluated. Before mealworms were introduced to the soil, the resporulated *M. anisopliae* granules or food granules were thoroughly mixed into the soil by carefully inverting the container several times. To remoisten the soil, 5 mL of sterile water was added to each tub. Twenty larval mealworms (mean 110 mg weight; 16 mm length and 1.6 mm wide) were added to each container, containers were loosely sealed with a lid and re-incubated at 25 °C, 65% RH in the dark in the same plant growth chamber. At 24 h post-incubation, a piece of fresh gold sweetpotato (~30 g) was supplied as food for the mealworms and the container lids were replaced with perforated lids to facilitate the aeration inside the containers. All containers were placed back into the plant growth chamber according to the previous design.

After the mealworms were exposed to the *M. anisopliae* or food granules, the mortality of the larval mealworms was assessed on daily basis for 14 days. During the assessment period, any dead mealworms were removed by using sterilized tweezers. To confirm EPF-induced mortality, dead insects were immediately surface sterilized and placed into a moist chamber (a Petri dish lined with a sterile filter paper moistened with 1 mL sterile water) at 25 °C to stimulate sporulation. The surface sterilization of mealworm cadavers was performed by placing the cadavers in 70% ethanol for few seconds, rinsing briefly in sterile water, dipping in 1% diluted sodium hypochlorite for 1 min, twice rinsing with sterile water and blotting dry on sterile filter paper.

### 2.9. Data Analysis

Data were tested for normal distribution and homogeneity of variance by using the Shapiro–Wilk and Levene test, respectively. Data obtained from ‘cultivation of soil microbes’ were firstly analyzed through two-way ANOVA to determine the interaction between two independent variables (soil levels and media types) in relation to the CFUs. Significant differences among field soil, pasteurized soil and fumigated soil in terms of CFU groups were calculated using ANOVA and their pairwise comparisons were further performed by using the Tukey test with 95% confidence. For the ‘resporulation assessment’ and ‘infectivity assessment,’ the data were analyzed using an ANOVA. For the experiment on ‘infectivity assessment,’ the mortality data were converted to percentages and then further corrected using Abbott’s formula [54]. The corrected data were subject to the ANOVA analysis.

## 3. Results

### 3.1. Cultivation of Soil Microbes

A significant interaction was found between soil type (field, pasteurized, fumigated 1 and fumigated 2) and media type (potato dextrose agar and nutrient agar) for colony-forming units (CFUs) (*p* = 0.001). The number of CFUs from field soil, pasteurized soil, fumigated 1 soil and fumigated 2 soils were significantly different from one another, irrespective of media type (*p* = 0.01) (Figure 3). For nutrient agar, the CFUs from field soil were significantly greater than those on pasteurized soil and fumigated 1 soil (*p* < 0.05), but not significantly different from that observed on fumigated 2 soils (*p* > 0.05). For potato dextrose agar, the number of CFUs in field soil was significantly greater than that on fumigated 2 soils (*p* < 0.05), but it was not significantly different to pasteurized soil and fumigated 1 soil (*p* > 0.05).

One *Metarhizium* spp. (putatively identified as *M. brunneum* after sequencing the translation elongation factor 1-alpha gene) was isolated from field soil using the mealworm-bait method on the 1fourth day post-exposure (Figure 4A,B), while at this time point, there was no evidence of any EPF found from pasteurized or fumigated soil using the mealworm-bait method. However, when examined approximately 30 days later, the same *Metarhizium*-like fungi was found across all soils (presenting as conidiated cadavers), suggesting the native EPF had recovered sufficiently from the treated soils to become infective again (Figure 4A–C).

### 3.2. Resporulation Assessment

Ten days after inoculation, there was substantial resporulation from the *M. anisopliae* granules (Figure 5A–C); however, there was no significant effect of soil treatment on the number of conidia from resporulated granules (*p* = 0.189). The resporulated conidia from the individual *M. anisopliae* granules on fumigated soil were calculated at 4.14 × 10^7^ (±2.17 × 10^6^) conidia per granule, 3.74 × 10^7^ (±8.88 × 10^5^) conidia per granule on field soil and 4.28 × 10^7^ (±2.67 × 10^6^) conidia per granule on pasteurized soil. Germination of resporulated conidia was significantly different (*p* = 0.0001) among the three soils treatments. *Metarhizium anisopliae* granules inoculated in field soil had shown the highest germination at 86% (± 2%), whereas the lowest germination at 63% (±2%) was recorded for the fungal granules applied on pasteurized soil. The conidia from the fungal granules applied in fumigated soil had 82% (±6%) germination (Figure 5D–F). Conidial germination of resporulated fungal granules between field soil and fumigated soil was statistically non-significant (*p* = 0.406), whereas the germination of conidia in pasteurized soil was significantly lower than that recorded from the field and fumigated soil (*p* = 0.0001) (Figure 6A). Germ-tube length was found significantly different among treatments (*p* = 0.0362) (Figure 6B). The resporulated conidia from the fumigated soil showed the longest germ-tube length, which was significantly different than that on pasteurized or field soils. Fungal granules inoculated on 3% water agar as a quality control substantially resporulated, yielding 2 × 10^7^ conidia from a single fungal granule, whereas food granules cultured on 3% water agar did not show any changes.

### 3.3. Infectivity Assessment

The assessment of mealworm mortality began four days following the insect exposure to soil with resporulated fungal granules and was observed daily. Mealworm mortality on pasteurized soil (80%) and fumigated soil (80%) with *M. anisopliae* granules was significantly greater than that in field soil (45%) on the fourth day post-inoculation (*p* = 0.001). Mealworm mortalities among field soil, fumigated soil and pasteurized soil inoculated with fungal granules were not significantly different from one another on the eighth day post-inoculation, where 100% mortality was recorded from all treatments (*p* > 0.05) (Figure 7). Substantial ecdysis (>90%) from larval mealworms exposed to soil containing fungal granules was evident on the fourth day post-inoculation, whereas there was not any ecdysis present on mealworms recovered from the soil with food granules (control) throughout the assessment. Those dead larval mealworms recovered at four and five days post-inoculation were deep brown, which led to mycosis and conidiation following the placement in a moist chamber (Figure 7A,B). At mortality assessment on the eighth day post-exposure on soil, the larval mealworms introduced on pasteurized soil with food granules had 5% mortality, whereas no death was observed in field soil and fumigated soil with food granules, despite substantial saprotroph growth shown on food granules. About 50% of larval mealworms exposed to soil containing food granules pupated during the 14-day bioassay.

## 4. Discussion

In our study, we examined the role of soil fumigation in reducing fungistasis and the resporulation and infectivity of calcium-alginate-encapsulated *M. anisopliae.* We used metham sodium that emits methyl isothiocyanates (MITC) upon contact with moist soil [55] and soil pasteurization as methods of soil fumigation. The principal purpose of soil fumigation is to control soil-borne plant parasitic fungi, nematodes, insect pests and weeds [33,39,40]. Soil fumigation has been linked to the suppression of a wide range of soil microbes, including eukaryotes and prokaryotes [33,49], but the suppressive effect is transient so that soil microbe numbers start to rebound once the chemical toxicity dissipates in the soil [56]. Thus, we aimed to achieve optimal saprophytic growth of *M. anisopliae* from food granules in soil when fungistasis was reduced by soil fumigation.

These results showed that fungal granules successfully resporulated on fumigated soil and the resulting resporulated granules in soil caused larval mealworm mortality. Although the number of conidia from the resporulated fungal granules was the same among three different soil treatments, conidial qualitative parameters, in terms of the physical appearance of resporulated fungal granules, conidial germination and length of germ tube of germinated conidia were significantly different. *Metarhizium anisopliae* granules inoculated in soil caused 100% mortality of larval mealworms within eight days of insect exposure to soil containing resporulated fungal granules, irrespective of soil treatment. When fungal granules were introduced into the soil, they substantially resporulated, for example, 3.74 × 10^7^ (±8.88 × 10^5^) conidia per fungal granule (originally 9 × 10^6^ conidia per granule were encapsulated) on field soil, producing an extensive number of viable conidia. The presence of this high conidial concentration with good viability resulted in the rapid death of all mealworms in our study. It is probable that the nutritional requirements of *M. anisopliae* were met in this study that allowed for prolific sporulation and saprophytic growth in soil, as shown in other studies [56,57,58]. As a facultative saprophyte, *M. anisopliae* can exploit externally supplied nutrients [59]; therefore, the fungal propagules, such as conidia or mycelia, are often fortified with nutritive additives to encourage successful saprophytic growth in soil [12,24].

The resporulated fungal granules inoculated on field soil did show microbial contamination based on the visual appearance compared to the granules on fumigated and pasteurized soil. Despite the microbial contamination on fungal granules, the food-fortified fungal granules successfully resporulated on field soil. Conducive environmental conditions in the laboratory, particularly soil moisture and temperature, and the placement of *M. anisopliae* granules on the soil surface could explain the success of *M. anisopliae* resporulation on field soil. *Metarhizium anisopliae* granules applied on the soil surface more readily access oxygen compared to granules inoculated deeper into the soil profile as the conidial germination and further growth of *M. anisopliae* significantly consumes oxygen (O_2_) [60]. The slow-growing fungal granules in deep soil profile are likely to be attacked by soil saprotrophs as the growth of soil saprotrophs, for example, species of *Mucor*, *Penicillium*, *Aspergillus* and *Fusarium* [61]. Moreover, the conidia extracted from the resporulated fungal granules on field soil demonstrated 86% (±2) conidial germination, which is significantly higher than those in pasteurized soil (62% ±2). Although resporulated fungal granules on field soil demonstrated the highest conidial germination, the length of conidial germ tube was found the lowest, as compared to other soil types. Background soil microbes on field soil could be a factor to counteract against conidial development, while insect mortality by *M. anisopliae* is generally attributed to the infective conidia with rapid conidial attachment and their further penetration on insect hosts [14].

From our observations, encapsulated *M. anisopliae* showed appropriate utilization of the nutrient additives for its resporulation, surpassing the saprophytic growth stage upon favorable abiotic factors, mainly soil moisture (>20% water holding capacity) and temperature (20–30 °C) [53]. In contrast, if such conditions do not exist during the fungal growth, the soil saprotrophs usually overtake the fungal growth, utilizing the food for themselves and reducing subsequent EPF fungal resporulation [31]. Thus, maintaining the optimal climatic conditions during the period of fungal resporulation in the soil is crucial but is a balancing act to ensure that conditions for the EPF are favored. For this reason, an idea of preconditioning of fungal granules before the inoculation in soil has been proposed to improve the chance of EPF inocula outcompeting other soil microbes [59]. There have been multiple attempts to reduce fungistasis to enhance the saprophytic growth of biological control agents using labile carbon sources, for example, sugars and amino acids, while fungistasis is likely to be formidable on carbon-deprived soil [30].

*Metarhizium* spp.-induced insect mortality on fumigated soil after soil treatment signified that the metham fumigation only caused a transient suppression of indigenous EPF. Similarly, one study also noted the revival of the indigenous population of mycorrhizal fungi on fumigated soil [62]. The application of metham as a soil fumigant does not decimate indigenous *Metarhizium* populations based on the observation from the above study, while a consortium of indigenous EPF and externally applied EPF can coexist [63]. The metham fumigation could have stimulated the resporulation of fungal granules by suppressing the deleterious soil microbes, which antagonized *M. anisopliae* in the field soil. The germination test from resporulated fungal granules signified that field soil and fumigated soil mediated the production of resporulated fungal granules, with significantly higher conidial germination than that for pasteurized soil, suggesting that the rate of microbial recovery in soil, especially fungal antagonists, is higher in pasteurized soil than fumigated soil. Furthermore, longer germ tubes of conidia from fumigated soil may suggest that the growth of germinated conidia was faster in treated soils, which might cause the rapid mortality of insects once the fungus penetrates the insect cuticle.

Encapsulated *Metarhizium* in calcium alginate may be more suitable than the directly exposed conidial formulation. For example, the *M. anisopliae* conidia exposed to brassica produced isothiocyanates failed to germinate, grow and infect host insects [64]. The co-application of biocontrol fungi, such as *Gliocladium roseum* and *Talaromyces flavus,* in the form of alginate granular formulation in combination with a sub-lethal dose of metham sodium into the soil suppressed the soil-borne fungal pathogen *Verticillium dahliae* [43]. This may imply that the combination of fungal granules and biofumigation might be possible because of the potential low deleterious effect of biofumigants against soil microbes when applied as a sub-lethal dose. The reduced deleterious effects of bio-fumigation on soil microbes have been shown in the study [38], which showed that the biocidal effect of biofumigation is transient and non-lethal as compared to metham sodium.

The greater level of mealworm mortality on fumigated (80% ± 3) and pasteurized soil (80% ± 2 on pasteurized soil) than that on field soil (45% ± 7) within four days of inoculation can be attributed to the viable fungal propagules resulting from the resporulation of fungal granules, as well as mealworms being a susceptible host insect to EPF. A similar result was also reported in a study [65], which reported the 100% mealworm mortality on the 6th day of inoculation carried out by dipping larval mealworms on the aqueous suspension (4 × 10^5^ to 4 × 10^6^ conidia mL^−1^) of *Metarhizium* spp. or *Beauveria* spp.

Our targeted host insect is wireworm, a soil insect infesting belowground plant parts, such as sweetpotato roots. Generally, soil insects require longer exposure time to be infected and killed by insect pathogens, for example, soil-applied *M. anisopliae* conidiated on millet grains at the rate of 10^7^ conidia g^−1^ soil resulted in the death of 50% of the population of the subterranean scarab larvae (*Adoryphorus couloni*) after 19 days in laboratory conditions [66]. Similarly, wireworms (*Agriotes* sp.) introduced into the soil with *M. anisopliae* as an aqueous formulation of 3.85 × 10^6^ conidia g^−1^ air-dried soil had 50% population mortality at 50–65 days post-inoculation in the laboratory [18]. A study demonstrated that soil insects, such as wireworms, establish a symbiosis with soil microbes, especially bacteria, for example, *Pantoea agglomerants* and *Pandoraea pnomenusa,* which safeguard wireworms from the infection of insect pathogens, such as *M. brunneum* [67]. In addition, insects were also found to maintain a host insect–actinomycete association on the insect exoskeleton, which acts as the first line of defense against insect pathogen infection [68]. For any insects, mortality is a function of fungal concentration and exposure time. For wireworm mortality, exposure for at least 48 h to *M. anisopliae* is estimated to be required [67], with a dose of at least 4 × 10^6^ conidia cm^−3^ soil [14]. Results from our current study show that the soil with resporulated fungal granules resulted in 100% mealworm mortalities within eight days of inoculation, suggesting that these resporulated fungal granules could also cause substantial wireworm damage, but unlike mealworms, wireworms may need longer exposure time to the fungal inocula to be infected [69]. The other factor that potentially impedes the fungal infection to wireworms is the behavioral constraints of some wireworms, where the fungal-contaminated area is avoided and wireworms migrate to non-contaminated areas [70]. This avoidance behavior was not observed in these experiments; in contrast, mealworms were observed to be actively feeding on the granules.

## 5. Conclusions

This study showed that fungal granules applied in the field, fumigated or pasteurized soil demonstrated substantial resporulation, despite the presence of significant differences in soil microbial numbers among these soils. However, qualitative attributes of fungal granules resporulating on fumigated soil were found to be better than those in the field and pasteurized soil in terms of conidial germination, length of the germ tube and cross-contamination by other soil saprophytic fungi. The complete mealworm mortality within eight days of exposure to resporulated fungal granules in soil showed that the resporulated fungal conidia from the granules (i.e., isolate QS155) in soil are highly pathogenic to insects. Moreover, fumigated soil (19 days after the soil fumigation) allowed the fungal granules to resporulate and, subsequently, the resultant fungal granules killed the insects, confirming that fungal granules can be applied in fumigated soil to enhance the efficacy of fungal resporulation but probably only when the biofumigant toxicity receded. Future studies can be directed towards the assessment of fungal sensitivity with metham fumigant in the glasshouse conditions and further expanded in the field, the impact of soil fumigation on the indigenous EPF and fate, persistence and infectivity of EPF in biofumigated soil.

## Figures and Tables

**Figure 1 jof-08-01114-f001:**
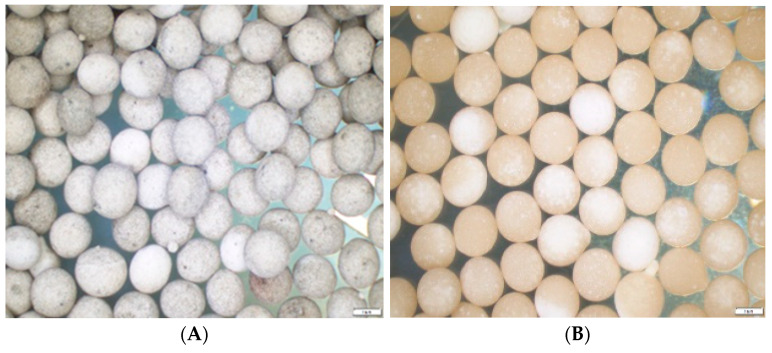
Fungal (*M. anisopliae*) granules (CAG_Ma+Cs+By_) (**A**) characterized with Ø = 3.5 mm, weight 25 mg per granule, 9 × 10^6^ conidia per granule and food (control) granules (CAG_Cs+By_) (**B**) as the control treatment.

**Figure 2 jof-08-01114-f002:**
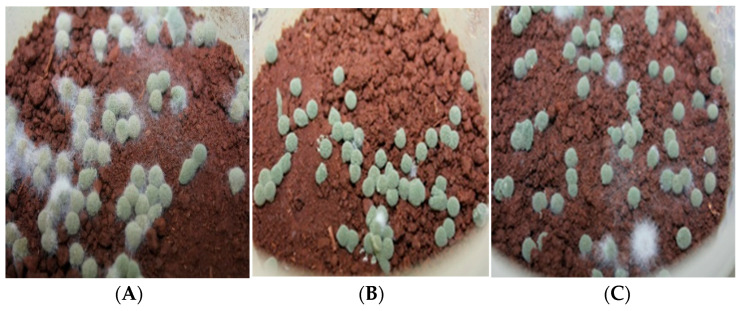
Resporulated *M. anisopliae* granules inoculated in field soil (**A**), fumigated soil (**B**) and pasteurized soil (**C**) 10 days post-incubation.

**Figure 3 jof-08-01114-f003:**
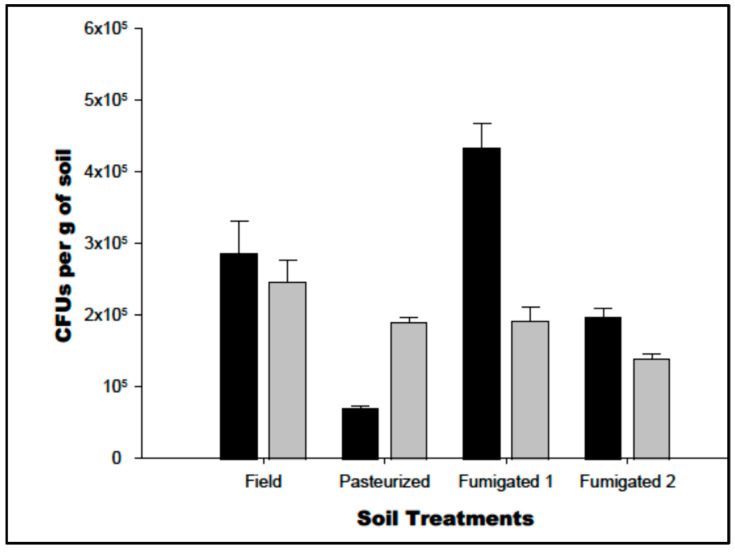
The colony-forming units (CFUs) present on standard nutrient agar (NA: black) or potato dextrose agar (PDA: Grey) media soil sourced from field soil, pasteurized soil, fumigated soil with soil sampled from the top fumigant-treated area (fumigated 1) and fumigated soil with soil sampled from homogenized fumigated soil (fumigated 2).

**Figure 4 jof-08-01114-f004:**
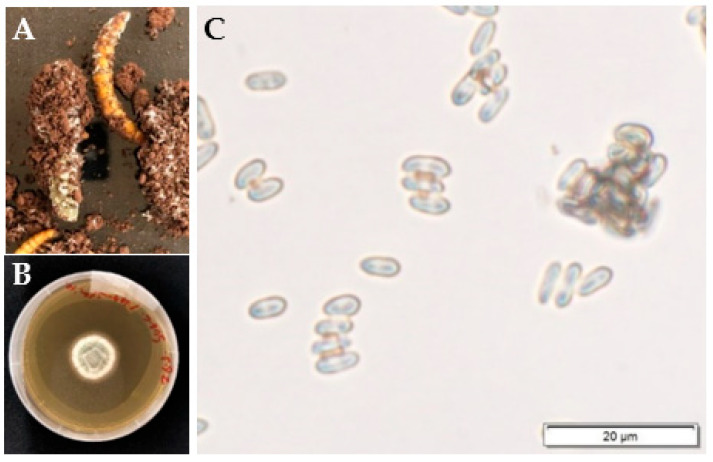
Sporulated cadavers were found in field soil samples sourced from the agricultural field at the USQ, Toowoomba (**A**); a fungal colony, suspected to be *Metarhizium brunneum* grown from a single colony isolated from an insect cadaver (**B**) and the fungal conidia obtained from the fungal colony (**C**).

**Figure 5 jof-08-01114-f005:**
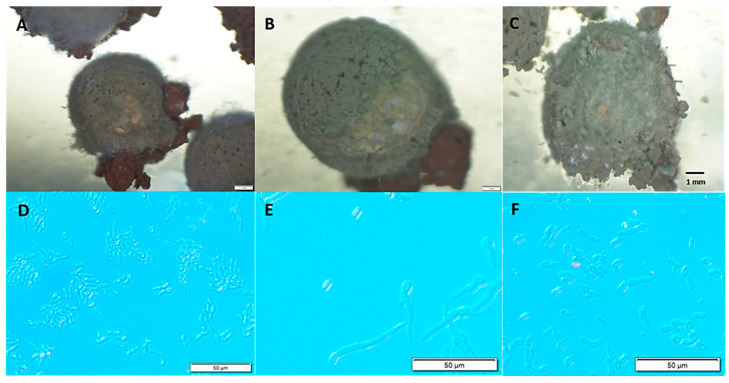
Resporulated fungal granules following 10 days incubation of fungal granules on field soil (**A**), fumigated soil (**B**) and pasteurized soil (**C**) and showing impact of soil treatments on conidial germination and germ-tube length (following a 14h incubation at 27 °C and a 12:12 h dark and light photoperiod) of the resporulated fungal granules grown on field soil (**D**), fumigated soil (**E**) and pasteurized soil (**F**).

**Figure 6 jof-08-01114-f006:**
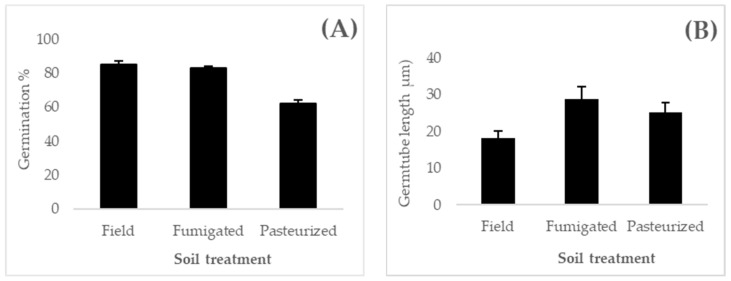
Percentage germination of extracted conidia from resporulated fungal granules applied on either field or fumigated or pasteurized soil for 10 days (*p <* 0.05) (**A**) and measurement of germ-tube length from the conidia extracted from the resporulated fungal granules (*p* < 0.05) (**B**).

**Figure 7 jof-08-01114-f007:**
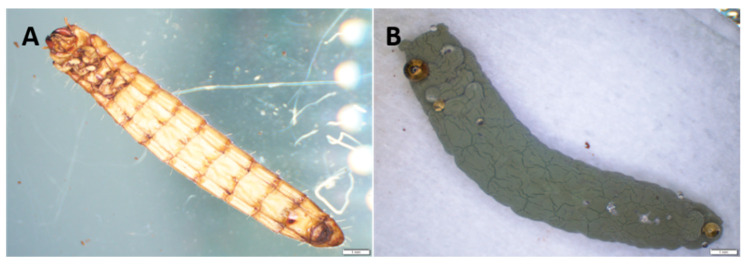
A dead larval mealworm recovered from fumigated soil containing resporulated fungal granules at 8 days post-inoculation with fungal inocula (**A**) a larval mealworm with profuse sporulation following placement in a moist chamber for 4 days (**B**).

## Data Availability

Not applicable.

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
