# Peer review of "Resporulation of Calcium Alginate Encapsulated Metarhizium anisopliae on Metham®-Fumigated Soil and Infectivity on Larvae of Tenebrio molitor"

_jof, 2022, doi:10.3390/jof8101114_

Round 1

Reviewer 1 Report

Shah et al. entitled “Resporulation of calcium alginate encapsulated Metarhizium anisopliae on Metham® fumigated soil and infectivity on larvae of Tenebrio molitor” reported that fungal granules can be applied in sweetpotato management. However, there are some issues that need to be demonstrated before it is published.

   Minor revision:

1.      It is better to state the reason why the infectivity was evaluated at 10 days but not other days.

Author Response

Feedback 1: It is better to state the reason why the infectivity was evaluated at 10 days but not other days.

Response: The infectivity assessment was done on daily basis for 14 days (please refer to page line 292), only the fungal granules used for the infectivity assessment was incubated for 10 days before it was undergone for the infectivity assessment.

Reviewer 2 Report

Review. Journal of Fungi.

Resporulation of calcium alginate encapsulated Metarhizium 2 anisopliae on Metham® fumigated soil and infectivity on larvae of Tenebrio molitor

By Shah et al.

This paper describes a well planned and carefully conducted study of the effect of prior fumigation or sterilisation of soil on resporulation of the entomopathogenic fungus Metarhizium anisopliae and subsequent pathogenic effect against the yellow mealworm, Tenebio molitor as a model for sweet potato pests, in particular wireworms.

I have no concerns about recommending the paper for publication following slight modification. There are some areas where the paper would benefit from another read through and some minor rewording to increase readability. I have noted some, instances of this, but by no means all. However, this is a small criticism, often a personal style issue and in most instances the meaning is clear.

Specific comments are listed below.

Line 13: Sweet potato is usually written as two words – multiple instances through the manuscript

Line 13 nutrient-fortified rather than “nutritive fortified”

Line 78 – insert ‘a’ before highly

Line 95 – typo – whilst.

Line 310 – based on Abbott’s formula – delete “the”

May want to specify here how Abbott’s formula was applied, presumably it was control within similar soil treatment and time point?

Line 314-316

Line 321 – “the number of CFUs in field soil were significantly greater than”-  should be was significantly greater than…

Line 320-322  For potato dextrose agar, the number of CFUs in field soil WAS significantly greater than THAT on fumigated 2 soils (P < 0.05), but it was not significantly different to pasteurized soil and fumigated 1 soil (P > 0.05).

Line 324-325. Given the frequent confusion surrounding M.a. strains, consider whether it would be worthwhile adding a supplementary data file here.

Line 359  “The re-sporulated conidia from the fumigated soil showed the longest germ length”. – maybe tube length rather than germ length?

Lines 344-346 – Maybe reword this caption.

Section 3.3. This section would be significantly improved by addition of a graphic for progressive mortalities in the different treatments and control – maybe relative survival curves

Lines 377-380. That is “Mealworm mortalities among field soil, fumigated soil and pasteurized soil inoculated with fungal granules were not significantly different from one another on the 8th day post-inoculation, where 100% mortality was recorded from all treatments (P = 0.087)”

The p value seems to conflict with the indication of 100% mortality in all treatments. The p value suggests that there were differences, albeit not significant – is this a wording issue?

Lines 380- 383. This sentence needs rewording. The intent of the sentence is not clear. As written it seems to  suggest that ecdysis is synonymous with/ leads to mortality?

Line 408 coma, not semi colon after 3rd soil.

Line 510-512. This conclusion is somewhat puzzling to me. The preceding paragraph gives an excellent discussion of characteristics/behaviours that have presumably evolved to protect soil inhabiting insects, including wireworms, from the effect of pathogenic soil fungi to protect them. However, the conclusion seems to be that false wireworms, which I don’t think have been mentioned anywhere else, may be more like mealworms. I would have thought that the conclusion from this section (and possibly in the suggestion for future studies in the last paragraph of the paper) should logically be that results could be different with soil-evolved insects (and further studies will be required to test this).

Author Response

Line 13: Sweet potato is usually written as two words – multiple instances through the manuscript

We disagree. The term ‘sweetpotato’ is globally accepted in the industry to separate it from the Irish potato. For more information, please follow the below provided references; 

Sweetpotato: One Word or Two? (cipotato.org)

Microsoft Word - SP info and history-1.doc (ucdavis.edu)

Australian Sweetpotato Growers Inc (aspg.com.au)

Line 13 nutrient-fortified rather than “nutritive fortified”

It has been corrected to nutrient-fortified as the reviewer suggested.

Line 78 – insert ‘a’ before highly

Added an article ‘a’ as the reviewer asked to do so.

Whilst

 The spelling mistake has been corrected

Line 310 – based on Abbott’s formula – delete “the”

May want to specify here how Abbott’s formula was applied, presumably it was control within similar soil treatment and time point?

‘The’ was deleted

Agreed with the reviewer that Abbott’s formula was applied between the treatment and the treatment specific control, for example fumigated soil with fungal granules (treatment) vs. fumigated soil with food granules (control).

Line 321 – “the number of CFUs in field soil were significantly greater than”-  should be was significantly greater than

 Changed to ‘was significantly greater than’ from ‘were significantly greater’.

Line 320-322  For potato dextrose agar, the number of CFUs in field soil WAS significantly greater than THAT on fumigated 2 soils (P < 0.05), but it was not significantly different to pasteurized soil and fumigated 1 soil (P > 0.05).

Replaced ‘those’ with ‘that’ as the reviewer asked to do so.

Line 324-325. Given the frequent confusion surrounding M.a. strains, consider whether it would be worthwhile adding a supplementary data file here.

A published paper will be provided as a supplementary evidence about the fungus, Metarhizium anisopliae.

Line 359  “The re-sporulated conidia from the fumigated soil showed the longest germ length”. – maybe tube length rather than germ length?

Yes, changed to ‘germ tube length’ from ‘germ length’.

Lines 365-368 – Maybe reword this caption.

Change made to ‘Resporulated fungal granules following 10 days incubation of fungal granules on field soil (A), fumigated soil (B), and pasteurized soil (C); and showing impact of soil treatments on conidial germination and germtube length (following a 14h incubation at 27 °C and a 12:12 h dark and light photoperiod) of the resporulated fungal granules grown on field soil (D), fumigated soil (E) and pasteurized soil (F)’.

Section 3.3: This section would be significantly improved by addition of a graphic for progressive mortalities in the different treatments and control – maybe relative survival curves

Insect mortality due to the fungal granules reached to 100% mortality on 8 days after the inoculation regardless of soil treatments. Thus, we decided that it is not worthwhile to show up this result in a graph as the result is insignificant among the treatments.

Lines 377-380. That is “Mealworm mortalities among field soil, fumigated soil and pasteurized soil inoculated with fungal granules were not significantly different from one another on the 8th day post-inoculation, where 100% mortality was recorded from all treatments (P = 0.087)”

The p value seems to conflict with the indication of 100% mortality in all treatments. The p value suggests that there were differences, albeit not significant – is this a wording issue?

There was a typo which is corrected by putting P value > 0.05.

Lines 380- 383. This sentence needs rewording. The intent of the sentence is not clear. As written it seems to  suggest that ecdysis is synonymous with/ leads to mortality?

Change was made as to ‘Substantial ecdysis (>90%) from larval mealworms exposed to soil containing fungal granules was evident on the 4th-day post-inoculation, whereas there was not any ecdysis present on mealworms recovered from the soil with food granules (control) throughout the assessment’.

Line 408 coma, not semi colon after 3rd soil.

Fixed, please refer to the relevant page.

Line 510-512. This conclusion is somewhat puzzling to me. The preceding paragraph gives an excellent discussion of characteristics/behaviours that have presumably evolved to protect soil inhabiting insects, including wireworms, from the effect of pathogenic soil fungi to protect them. However, the conclusion seems to be that false wireworms, which I don’t think have been mentioned anywhere else, may be more like mealworms. I would have thought that the conclusion from this section (and possibly in the suggestion for future studies in the last paragraph of the paper) should logically be that results could be different with soil-evolved insects (and further studies will be required to test this).

As the reviewer suggested, we removed any sentence or words reducing the readability, for example false wireworms. Rather, the conclusion was made much relevant as follows: This study showed that fungal granules applied in the field, fumigated, or pasteurized soil demonstrated substantial resporulation, despite the presence of significant differences of soil microbial numbers among these soils. But qualitative attributes of fungal granules resporulating on fumigated soil were found to be better than those in the field and pasteurized soil in terms of conidial germination, length of the germ tube, and cross-contamination by other soil saprophytic fungi. The complete mealworm mortality within eight days of exposure to resporulated fungal granules in soil showed that the resporulated fungal conidia from the granules (i.e., isolate QS155) in soil are highly pathogenic to insects. Moreover, fumigated soil (19 days after the soil fumigation) allowed the fungal granules to resporulate and subsequently, the resultant fungal granules killed the insects, confirming that fungal granules can be applied in fumigated soil to enhance the efficacy of fungal resporulation but probably only when the biofumigant toxicity receded. Future studies can be directed towards the assessment of fungal sensitivity with metham fumigant in the glasshouse conditions and further expanded in the field, the impact of soil fumigation on the indigenous EPF, and fate, persistence, and infectivity of EPF in biofumigated soil.

Reviewer 3 Report

In this manuscript, the fungal resporulation was evaluated in Metham®-fumigated soil and the infectivity of the resulting granule sporulation on mealworm. And results showed that the fungal granules can be applied in sweetpotato fields in conjunction with soil fumigation. I provided some comments for the authors to consider as outlined below.

1. Abstract need to be extended by using more significant results.

2. Line 37-38: Please rewrite the sentence to connect above-mentioned sentence.

3. Line 72, Line 109, et al: The order of references should be corrected throughout the manuscript.

4. The results should be shown more clarifying and describing. And the results of Fig. 6 should be statistical analysis.

5. More detail of infectivity assessment should be present, for example, the daily mortality rate; the mean lethal time (LT50) values, et al.

Author Response

Abstract need to be extended by using more significant results.

The abstract has been amended as per the reviewer comment to as ‘The fungal granules grown on different soil treatments (fumigated, field and pasteurized soil) resporulated profusely (for example, 4.14 × 107 (± 2.17 × 106) conidia per granule on fumigated soil), but the resporulation was not significantly different among the three soil treatments. However, the conidial germination of the resporulated granules on fumigated soil was > 80%, which was significantly higher than those on pasteurized soil or field soil. The resporulated fungal granules were highly infective, causing 100% insect mortality 9 days after the inoculation, regardless of soil treatments. The results from this research show that the fungal granules applied to soils could be an infective inoculant in sweetpotato fields in conjunction with soil fumigation. Additional field studies are required to validate these results and to demonstrate integration with current farming practices’.

Line 37-38: Please rewrite the sentence to connect above-mentioned sentence.

Change has been made.

The results should be shown more clarifying and describing. And the results of Fig. 6 should be statistical analysis.

Following change has been made ‘Percentage germination of extracted conidia from resporulated fungal granules applied on either field or fumigated or pasteurized soil for 10 days (P < 0.05) (A) and measurement of germ tube length from the conidia extracted from the resporulated fungal granules (P < 0.05) (B)’.

More detail of infectivity assessment should be present, for example, the daily mortality rate; the mean lethal time (LT50) values, et al.

We decided to leave as it is because amendment may not significantly improve, as we did not use multiple doses.